# Distinct Roles of Estrogen Receptors in the Regulation of Vitellogenin Expression in Orange-Spotted Grouper (*Epinephelus coioides*)

**DOI:** 10.3390/ijms23158632

**Published:** 2022-08-03

**Authors:** Zhifeng Ye, Tingting Zhao, Qianhao Wei, Haoran Lin, Yong Zhang, Shuisheng Li

**Affiliations:** 1State Key Laboratory of Biocontrol and School of Life Sciences, Southern Marine Science and Engineering Guangdong Laboratory (Zhuhai), Guangdong Provincial Key Laboratory for Aquatic Economic Animals and Guangdong Provincial Engineering Technology Research Center for Healthy Breeding of Important Economic Fish, Sun Yat-Sen University, Guangzhou 510275, China; yezhf3@foxmail.com (Z.Y.); tingzhao512@163.com (T.Z.); weiqh3@mail2.sysu.edu.cn (Q.W.); lsslhr@mail.sysu.edu.cn (H.L.); 2Laboratory for Marine Fisheries Science and Food Production Processes, Qingdao National Laboratory for Marine Science and Technology, Qingdao 266373, China

**Keywords:** vitellogenin, estrogen, estrogen receptor, hepatocyte

## Abstract

During their breeding season, estrogen induces vitellogenin (VTG) production in the liver of teleost fish through estrogen receptors (ERs) that support oocyte vitellogenesis. There are at least three ER subtypes in teleost fish, but their roles in mediating E_2_-induced VTG expression have yet to be ascertained. In this study, we investigated the expression of *vtgs* and *ers* in the liver of orange-spotted grouper (*Epinephelus coioides*). Their expression levels were significantly increased in the breeding season and were upregulated by an estradiol (E_2_) injection in female fish, except for the expression of *erβ1*. The upregulation of *vtgs*, *erα* and *erβ2* by E_2_ was also observed in primary hepatocytes, but these stimulatory effects could be abolished by ER antagonist ICI182780 treatment. Subsequent studies showed that ERβ antagonist Cyclofenil downregulated the E_2_-induced expression of *vtg*, *erα*, and *erβ2*, while the ERβ agonist DPN simulated their expression. Knockdown of *erβ2* by siRNA further confirmed that ERβ2 mediated the E_2_-induced expression of *vtgs* and *erα*. To reveal the mechanism of ERβ2 in the regulation of *erα* expression, the *erα* promoter was cloned, and its activity was examined in cells. E_2_ treatment simulated the activity of the *erα* promoter in the presence of ERβ2. Deletions and site-directed mutations showed that the E_2_ up-regulated transcriptional activity of *erα* occurs through a classical half-estrogen response element- (ERE) dependent pathway. This study reveals the roles of ER subtypes in VTG expression in orange-spotted grouper and provides a possible explanation for the rapid and efficient VTG production in this species during the breeding season.

## 1. Introduction

Vitellogenin (VTG), the precursor of the yolk protein in egg-bearing vertebrates, is essential for oogenesis, embryonic development, and larval survival. During the reproductive season, Vtg in the female liver is synthesized by estrogen stimulation and secreted into the bloodstream. Plasma Vtg is taken up by Vtg receptors on the surfaces of oocytes and cleaved into smaller yolk proteins by cathepsin [1]. Vitellogenesis is the primary phase of oocyte development and consumes a significant amount of nutrients and energy [2]. An insufficient vitellogenin uptake by oocytes leads to incomplete larval development and the higher mortality of eggs [3]. In female zebrafish, the knockout of *vtg* genes results in a dramatic drop in egg fertilization rates and offspring survival [4]. Therefore, substantial Vtg production is critical to the reproductive success of teleosts.

Piscine VTG are divided into three categories, namely, VTGAA, VTGAB, and VTGC, which are encoded by *vtgaa*, *vtgab*, and *vtgc*, respectively [5]. VTG protein could be rapidly produced in the liver after exogenous estrogen treatment [6]. Estrogen exerts its actions via activating estrogen receptors (ERs) [7]. At present, at least three estrogen receptor subtypes (ERα, ERβ1, and ERβ2) have been identified in fish species [8,9]. ERα is considered to be responsible for mediating the estrogen action in VTG synthesis in fish [10,11,12,13], as evidence has shown that the hepatic ERα expression was increased in the breeding season and was dramatically upregulated by E_2_ treatment [14,15,16].

Other piscine ER subtypes are also expressed in the liver besides ERα [17,18]. Whether they are involved in VTG production and the regulatory relationships between ER subtypes in this process attract attention. An intraperitoneal injection of E_2_ could upregulate the mRNA expression of *erβ1* and *erβ2* in the liver of goldfish [19]. ERβ selective agonist DPN increased VTG expression in rainbow trout hepatocytes in a dose-dependent manner [20]. A knockdown of *erβ1* or *erβ2* using specific siRNA in goldfish primary hepatocytes significantly reduced the E_2_-induced mRNA expression of *vtg* and *erα* [21]. Research on zebrafish embryos showed that the knockdown of *erβs* via specific morpholinos could suppress the estradiol induction of *vtg* and *erα* mRNA expression [22]. These studies indicate that ERβ subtypes play important roles in the vitellogenesis of fish. It is proposed that ERβ subtypes mediate estrogen signals to promote *erα* expression, which then sensitize the hepatocytes to further E_2_ stimulation and prepare them for vitellogenesis [21]. However, a systematic study in fish revealing the interplay between ER subtypes in VTG production and the regulatory mechanism is lacking. Moreover, further studies in different fish species with different reproductive modes are required to evaluate the significance of ER subtypes in this physiological process and reveal other important aspects of ER subtypes regarding the regulation of VTG production.

Groupers are marine benthic fish, belonging to the subfamily Epinephelinae (Teleoste: Serranidae), which contains more than 160 species in 15 genera [23]. Most groupers are protogynous hermaphrodites and are of significant economic value. According to the statistics from the Food and Agriculture Organization of the United Nations (FAO), global grouper production reached about 234,828 tons in 2019. Groupers generally enter the breeding season from the end of spring to the early autumn, and this process is mainly affected by water temperature [24]. During the breeding period, females carry 70,000 to 1,000,000 ova, depending on their size [24]. Due to large-scale farming in the Asia-Pacific region, the fertilized eggs of groupers have been in short supply over the past decade. Vitellogenesis is a crucial process for oocyte development. However, there are few studies on vitellogenesis, particularly on VTG regulation in groupers. In this study, using the orange-spotted grouper (*Epinephelus coioides*) as an experimental model, we aimed to evaluate the functional role of ER subtypes in the regulation of VTG production and to reveal their regulatory relationships. Firstly, the expression patterns of *ers* and *vtgs* in the liver during the breeding and non-breeding seasons were examined. Secondly, the regulation of *ers* and *vtgs* expression by E_2_, ER antagonists, and ER agonists was further investigated. Finally, the regulatory relationships between ER subtypes and the molecular mechanism of ERβ2 regulation towards *erα* expression were studied. Our study will not only characterize the roles of ER subtypes in the regulation of VTG production, but also reveal the transcriptional regulatory mechanism of how ERβ2 regulates *erα* expression in fish at the molecular level.

## 2. Results

### 2.1. Expression Profiles of vtgs and ers in the Liver of Orange-Spotted Grouper

To investigate the expression profiles of *vtgs* and *ers* in the liver during vitellogenesis, a histological analysis was firstly applied to examine the ovarian developmental status of each fish. The male groupers used as controls showed normal spermatogenesis in the testis (Figure 1A). Primary oocytes were present in the ovaries of non-breeding females (Figure 1B), while the females in the breeding season showed vitellogenesis in the ovaries with vitellogenic oocytes (Figure 1C). The results showed that females in the breeding season had significantly higher VTG protein and mRNA levels compared to females in the non-breeding season and males (Figure 1D–G). All three *er* genes were expressed in the livers of the groupers. Their mRNA levels in females were also obviously upregulated in the breeding season (Figure 1H–J). Evidently, *erα* expression showed a dramatic increase in the breeding season compared to the non-breeding season.

### 2.2. Effects of E_2_ on the Hepatic Expression of vtgs and ers in Juvenile Female Orange-Spotted Grouper

To investigate the relationship between *ers* and *vtgs* in vivo, the juvenile females received an intraperitoneal injection with a dose of 5 mg/kg E_2_. The results showed that exogenous estrogen dramatically increased the protein and mRNA levels of *vtgs* in the liver compared to the control group (Figure 2A–D). However, the *er* genes in the liver showed distinct expression patterns in response to E_2_ treatment. A remarkable increase in *erα* mRNA levels was observed, whereas a significant decrease in *erβ1* mRNA levels was found after the E_2_ administration (Figure 2E,F). The *erβ2* mRNA levels were slightly but significantly elevated following the E_2_ injection (Figure 2G).

Moreover, a primary hepatocyte culture system was established using the livers of the juvenile females, and the cells were incubated with different concentrations of E_2_ (from 0 to 1 μM) for 24 h. The results showed that E_2_ treatment significantly stimulated the protein and mRNA expression of *vtgs* in a dose-dependent manner (Figure 3A–D), and evoked the dose-dependent expression of *erα* and *erβ2* in the hepatocytes (Figure 3E,G). However, a dose-dependent decrease in *erβ1* expression was observed with increasing concentrations of E_2_ treatment (Figure 3F).

### 2.3. Effects of ER Antagonists or Agonist on the Expression of vtgs and ers in Primary Hepatocytes

To reveal the roles of different ERs towards the regulation of *vtg* expression, ER antagonists and one agonist were applied to treat the primary hepatocytes. These drugs altered the *er* expression and ultimately modulated the *vtg* expression. As shown in Figure 4, ERα and ERβ antagonist ICI182780 markedly suppressed the increasing protein and mRNA expression of *vtgs* induced by E_2_ in the hepatocytes and abolished the stimulatory effects of E_2_ on the expression of *erα* and *erβ2* but had no effect on the reduced expression of *erβ1* caused by E_2_.

Cyclofenil, an Erβ-specific antagonist, significantly attenuated the stimulatory effects of E_2_ on the expression of *vtgs*, *erα*, and *erβ2*, but could not change the *erβ1* expression patterns following co-incubation with E_2_ (Figure 5).

DPN, an ERβ specific agonist, stimulated the protein and mRNA expression of *vtgs* effectively, and significantly upregulated the mRNA expression of three *ers* in hepatocytes without the presence of E_2_ (Figure 6).

### 2.4. Effects of erβ2 Knockdown by siRNA on the Expression of vtgs and ers in Primary Hepatocytes

To verify whether ERβ2 could regulate the expression of *erα*, we established a transfection system that transfers the siRNA into primary grouper hepatocytes using an electroporation method. Flow cytometry was used to evaluate the transfection efficiency in the primary hepatocytes via electroporation. The fluorescence intensity of the hepatocytes in the control group and the siRNA-NC-Cy3 (NC) group was analyzed by flow cytometry (Appendix A), and the P1 range was defined as the strong fluorescence range. As shown in Appendix A, in the P1 range, the cell count of the NC group was significantly higher than that of the control group, indicating that the siRNA was effectively transmitted into the primary hepatocytes. To test the hypothesis wherein E_2_-induced *erβ2* expression would promote the *erα* expression and thus accelerate the *vtg* expression, two distinct siRNAs (Si217 and Si353) against *erβ2* were separately transfected into the hepatocytes through electroporation. As shown in Figure 7, both siRNAs could significantly reduce the E_2_-induced *erβ2* expression, resulting in an about 32% and 47% reduction compared to the E_2_ treatment group. As expected, the expression levels of the *vtgs* and *erα* simulated by E_2_ were significantly downregulated in the hepatocytes, and the *erβ1* expression was not affected by *erβ2* knockdown (Figure 7).

### 2.5. Estrogen Stimulation on erα Promoter Activity via ERβ2

Our results indicate that ERβ2 could mediate E2 action and then regulate the *erα* expression, but the transcriptional regulatory mechanism is unknown. Therefore, to reveal the mechanism of E_2_ in the regulation of *erα* expression via ERβ2, the 5′-flanking region of the *erα* gene was isolated. As shown in Appendix A, the cloned sequence upstream of the TSS of *erα* is 1434 bp in length, containing three half-estrogen response elements (ERE) and several transcription factor binding motifs, such as Activator protein 1 (Ap1), Specificity proteins 1 (Sp1), and cAMP response element binding protein (CREB). Whether the *erα* promoter activity was regulated by E_2_ and ERβ2 was further assessed in HEK293T cells. The results showed that E_2_ treatment significantly increased the *erα* promoter activity in the cells co-transfected with the ERβ2 (Figure 8A). The promoter activity was significantly elevated by E_2_ at 0.1 and 1 μM in the presence of ERβ2 (Figure 8B). To identify the E_2_ response region of the *erα* promoter, deletion analyses were further performed. The E_2_ treatment significantly increased the activity of the wild-type *erα* promoter in the presence of ERβ2, and the deletion of the promoter to position −1057 did not eliminate the E_2_ stimulatory effect (Figure 8C). However, deleting the promoter to position −825 could abolish the E_2_-induced promoter activity (Figure 8C), indicating that this region is responsible for mediating the E_2_ action towards triggering the transcriptional activity of the *erα* promoter through ERβ2. As the proximal −825 bp region of the *erα* promoter contains a putative Sp1 binding site and two putative half-ERE sites (marked by #1, #2) (Figure 8D), site-directed mutagenesis analyses were carried out. The mutation of the Sp1 binding site (upstream of TSS from −953 to −944 bp) or ERE#1 site (upstream of TSS from −900 to −895 bp) did not change the E_2_-induced promoter activity in the presence of ERβ2; however, the stimulatory effect was abolished when ERE#2 (upstream of TSS from −870 to −865 bp) was mutated (Figure 8D), indicating that ERE#2 is responsible for the E_2_/ ERβ2-induced expression of *erα*.

## 3. Discussion

The orange-spotted grouper is a seasonally spawning fish. In the breeding season, the serum estrogen levels in females are significantly elevated [25], but the pathway of estrogen in the regulation of the VTG product in the liver via ERs remains unclear in this species. In the present study, our results showed that (1) *erα* was abundantly expressed in the liver in the breeding season, (2) hepatic *erα* expression was more sensitive to E_2_ treatment than *erβs* in vivo and in vitro, (3) and that the E_2_-induced expression of VTG and *erα* was almost completely inhibited after the ICI182780 treatment, suggesting that hepatic ERα is mainly responsible for mediating the action of estrogen to promote the VTG production. These results are consistent with the findings in other fishes [26].

For *erβ1*, its expression in females was also significantly increased in the liver in the breeding season compared to non-breeding season. However, the E_2_ treatment reduced *erβ1* expression in vivo and in vitro, and this inhibitory effect was not abolished by ICI182780 and Cyclofenil. These results indicate that the elevated expression of *erβ1* in the breeding season may not be induced by estrogen. A study also found that hepatic *erβ1* expression was markedly decreased in zebrafish after 48 h of E_2_ exposure [9]. However, the hepatic *erβ1* expression level varied greatly in response to E_2_ in goldfish. Some studies observed an induction of hepatic *erβ1* expression by E_2_ in juvenile and female goldfish, while others found a downregulation of *erβ1* expression in males after E_2_ implantation, suggesting that Erβ1 possesses hepatic functions in teleosts, but its role in the regulation of VTG production is unclear.

Compared with other ER subtypes, *erβ2* showed different expression patterns. Its expression was significantly increased in the breeding season and after E_2_ treatments, but not as dramatically as *erα*. The ER antagonists, especially the Cyclofenil (a specific antagonist of ERβ), significantly attenuated the upregulation of *erβ2* mRNA expression induced by E_2_. These data indicate that hepatic ERβ2 expression is regulated by estrogen. Interestingly, Cyclofenil also downregulated the increased expression of the *erα* and *vtgs* induced by E_2_, but DPN upregulated their expression. As the hepatic *erβ1* expression was downregulated by E_2_ and was not sensitive to Cyclofenil, the actions of Cyclofenil and DPN on the expression of *erα* and *vtgs* may possibly occur via ERβ2. To verify whether ERβ2 could regulate the expression of the *erα* and *vtgs*, we established a transfection system that transfers the siRNA into primary grouper hepatocytes using an electroporation method. Two siRNAs against *erβ2* were employed to specifically knock down the hepatic *erβ2* expression, and a significant decrease in the E_2_-induced expression of *erα* and *vtgs* in the hepatocytes was also observed. However, unlike ICI182780, neither Cyclofenil nor siRNA knockdown could fully abolish the E_2_ induction of *erα* and *vtgs* expression, and the DPN-mediated upregulation of *erα* and *vtgs* expression was not as drastic as the E_2_ treatment, suggesting that ERβ2 may partially contribute to the estrogenic regulation of *erα* and *vtgs* expression in grouper.

The results above showed that ERβ2 could promote the hepatic *erα* expression by mediating the estrogen action, but the molecular mechanism remains unclear. At present, it is clear that estrogen’s regulation of downstream gene expression occurs through the E_2_-ER-ERE pathway. The E_2_-ER dimer complexes act on the ERE site in the promoter to initiate or repress gene expression [27,28]. Studies have shown that there are several ERE sites in the *erα* promoter region of rainbow trout and zebrafish [28]. However, there is no complete ERE site in grouper’ *erα* promoter, but three 1/2 ERE sites, and some transcription binding sites that could mediate the estrogen signal, including one Sp1 binding site, nine Ap1 binding sites, and one CRBE binding site, were observed. Subsequent experiments demonstrated that E_2_ significantly enhanced the grouper *erα* promoter activities in the presence of ERβ2 in HEK293T cells. Deletion and site-directed mutagenesis analysis indicated that the ERE#2 site is required to maintain the E_2_ induction of *erα* promoter activities, suggesting that ERβ2 could modulate *erα* transcriptional activity through a half-ERE-dependent classical mechanism in grouper. Our previous study on grouper has shown that the half-ERE site in the gene promotor could initiate the estrogen response [29].

The present data suggest that ERβ2 is involved, directly and indirectly, in the regulation of VTG production in grouper. As promoters of *vtg* genes contain E_2_ response sites, ERβ2 may bind to these sites and initiate the *vtg* expression in the hepatocytes directly. In addition, some studies proposed that ERα and ERβ2 may interact as heterodimers to drive the *vtg* expression [7]. However, this hypothesis requires further evidence in fish species. Indirectly, E_2_-activated ERβ2 enhances the hepatic *erα* expression, and then ER*α* promotes VTG synthesis, as our results demonstrate that E_2_ could regulate the *erα* expression in the presence of ERβ2. During the breeding season, VTG is produced rapidly and efficiently in the female bloodstocks. The collaboration between ERα and ERβ subtypes may accelerate this process. In goldfish, a model for E_2_-mediated vitellogenesis was proposed: both ERβ subtypes can induce some vitellogenesis and are necessary for the normal E_2_ induction of *erα* expression. The increased *er*α expression would sensitize the hepatocyte to further E_2_ stimulation, and thus promote VTG production in the breeding season. This model may not be entirely suitable for grouper. Our study shows that grouper ERβ1 may not be involved in the regulation of VTG production. Although blocking ERβ2 action and the knockdown of its expression could result in the reduction of erα expression, whether ERβ2 is required for the E_2_-mediated upregulation of *erα* expression is uncertain and awaits further study. Therefore, the roles of ER subtypes, especially ERβ receptors, in mediating estrogen action may be different across fish species, suggesting that fish have evolved different mechanisms to regulate VTG synthesis.

In summary, we investigated the roles of ER subtypes towards VTG production in orange-spotted grouper. The results revealed that the physiological actions of estrogen on VTG production are primarily mediated by ERα, and that ERβ2 may play a role in promoting the E_2_-mediated upregulation of *erα* and *vtg* expression. This study will lead to a better understanding of vitellogenesis and paves the way for improving the oocyte quality of groupers.

## 4. Materials and Methods

### 4.1. Animal and Sample Collection

Eighteen adult orange-spotted groupers (5-years old; body weight: 3.21~4.46 kg; standard length: 46.93~60.16 cm) were obtained from the Hainan Chenhai Fisheries Co., Ltd., in Sanya (Hainan Province, China). The gonadal status of fish was extracted with a catheter and syringe, immediately fixed in Bouin’s fluid, and checked by histological examination. The methods were carried out as described by Tang et al. [30]. Livers from non-breeding females (*n* = 6) were collected in February 2021, and livers from males (*n* = 6) and breeding females (*n* = 6) were collected in April 2021. Fish were anesthetized with MS-222 (Sigma) and dissected. The samples were frozen in liquid nitrogen and stored at −80 °C for RNA extraction. Juvenile female groupers (1.5-years old; body weight: 0.70~1.01 kg; standard length: 18.35~22.18 cm) used for intraperitoneal injection and primary hepatocyte culture were provided by Yuxiangzi Aquatic Industry Co., Ltd. in Yangjiang (Guangdong Province, China). All animal experiments were conducted in accordance with the guidelines and approval of the Animal Research and Ethics Committees of the Sun Yat-sen University.

### 4.2. Chemicals

17β-Estradiol (E_2_) (CAS 50-28-2; purity ≥ 98%) was purchased from Sigma. ICI182780 (CAS 129453-61-8; purity ≥ 99%) was purchased from Abcam. Cyclofenil (CAS 2624-43-3; purity ≥ 99%) and DPN (CAS 1428-67-7; purity ≥ 99%) were purchased from Tocris. E_2_, ICI182780, Cyclofenil, and DPN were dissolved in DMSO at a final stock concentration of 10 μM.

### 4.3. Intraperitoneal Injection Experiment

Sixteen healthy juvenile female groupers (1.5-years old; body weight: 0.70~1.01 kg; standard length:18.35~22.18 cm) were randomly divided into two groups, and kept in two 3000 L tanks with continuous seawater and air supply. Groupers were acclimatized for a week before the experiment. The fish were exposed to natural photoperiods and water temperature (25.5–28.2 °C) and fed with commercial feed once a day. E_2_ was dissolved in DMSO, and then diluted in physiological saline solution. The fish in experimental group were intraperitoneally injected with E_2_ solution (5 mg/kg body weight). The physiological saline injected group was used as control. The livers were collected at 24 h after injection. The mRNA levels of *vtgs* and *ers* were quantified by Quantitative real-time PCR. The content of VTG protein in the liver was measured by using a VTG ELISA kit (Cusabio, Wuhan, China) following manufacturer’s protocol.

### 4.4. Isolation, Primary Culture, and Static Incubation of Grouper Hepatocytes

Livers from six juvenile female groupers were excised and washed three times in ice-cold Ca^2+^/Mg^2+^-free HBSS (Gibco, Carlsbad, CA, USA) containing 1% antibiotic-antimycotic (Gibco). The tissue fragments were further diced to about 1 mm in thickness using a razor blade and washed three times in Ca^2+^/Mg^2+^-free HBSS containing 1% antibiotic-antimycotic at room temperature. Subsequently, slices were digested with trypsin-EDTA (0.25%) (Gibco) at 28 °C for 20 min, then were mechanically dispersed into individual cells by gentle pipetting and filtered through a 30 μm sterilized nylon mesh. Cells were harvested by centrifugation (30× *g* for 5 min, twice) and were resuspended in L-15 medium (Gibco) containing 1% antibiotic-antimycotic and 10% FBS (Gibco). The cell viability test and counting were performed using a trypan blue exclusion test. Only cells with >95% viability were used for subsequent experiments. The number of hepatocytes was adjusted to approximately 1.5 × 10^6^ cells/mL/well and seeded as a monolayer into 24 well-plates precoated with 5 µg/mL PEI (Sigma, Saint Louis, MO, USA). After preincubation at 28 °C for 24 h, the medium was removed and replaced with a fresh medium containing the test drugs (E_2_, ER antagonists and agonist). In the E_2_ treatment experiment, the doses of E_2_ were 0.001, 0.01, 0.1, and 1 µM. In ER antagonists’ experiments, the dose of E_2_ was 0.01 µM, and 1 µM of ICI182780 or Cyclofenil was used. In ER agonist’s experiment, 1 µM DPN was added into the medium. Each experiment had three groups (4 wells per group). After 24 h incubation, the culture media were collected for VTG protein assay, and cells were harvested for gene expression analysis.

### 4.5. Knockdown Experiment

Small interfering RNAs (siRNAs) (Table 1) were synthesized by GenePharma according to the sequence of *erβ2* (GenBank Accession No GU721078.1). Transfection experiment was performed on an electroporation instrument NEPA21 (NEPAGENE, Chiba, Japan). Primary hepatocytes (1.5 × 10^7^ cells per mL) were suspended in the opti-MEM and mixed with siRNA at room temperature for 10 min. The mixture was transferred into electrode chamber and subjected to electroporation. The program was 195 V, 2.5 ms, 2 times, and 50 ms interval for poring pulse, and 20 V, 50 ms, 5 times, and a 50 ms interval for transfer pulse. After electroporation, the mixture was immediately transferred to L-15 medium containing 10% FBS in a 24-well plate. After preincubation at 28 °C for 24 h, the medium was removed and replaced with a fresh medium containing E_2_. After 24 h incubation, cells were harvested for gene expression analysis. The electroporation experiment was performed in triplicate.

To analyze the transfection efficiency, siRNA-NC-Cy_3_ (Cy_3_-labeled negative control siRNA) was transfected into primary hepatocytes by electroporation according to the protocol above. After 24 h incubation, the fluorescence value was detected by Beckman Cytoflex flow cytometry (Beckman Coulter Inc., Brea, CA, USA), and analyzed by CytExpert software (Beckman Coulter Inc., Brea, CA, USA).

### 4.6. Quantitative Real-Time PCR

To examine the mRNA levels of *vtg*s (*vtgaa, vtgab*, and *vtgc*) and *ers* (*erα, erβ1*, and *erβ2*), total RNA was extracted with Trizol reagent (Invitrogen, Carlsbad, CA, USA) and was reverse transcribed with a ReverTra Ace qPCR-RT Kit (Toyobo, Osaka, Japan) following the manufacturer’s protocol. Quantitative real-time PCR reactions were performed on a Roche Light-Cycler 480 real-time PCR system using SYBR Green I Master (Roche, Basel, Switzerland) according to the manufacturer’s protocol. Briefly, 500 ng template, 5 µL SYBR Green I Master, and 0.2 µL forward (reverse) primer (10 mM) were mixed in a tube and supplemented with water to 10 µL. The program was as follows: denaturation at 95 °C for 10 min, followed by 40 amplification cycles of 95 °C for 10 s, 58 °C for 20 s, and 72 °C for 20 s. After amplification, melting curve analysis was carried out to confirm the specificity of amplification. Fluorescence signals were converted to threshold cycle (Ct) values. *Beta-actin* was used as reference gene to normalize the expression values. The relative mRNA levels were calculated by using 2^−ΔΔCT^ method. The sequences of gene-specific primers are listed in Table 2.

### 4.7. Plasmid Construction and In Silico Analysis of Promoter

Using a SMARTer RACE cDNA Amplification Kit (Takara, Osaka, Japan), the transcriptional start site (TSS) of *erα* gene were confirmed. The 5′-flanking region (1434 bp) upstream of the transcriptional start site was cloned from the genome of orange-spotted grouper. The transcription factor binding sites in the promoter region were predicted using a virtual database online (http://alggen.lsi.upc.es/cgi-bin/promo_v3/promo/promoinit.cgi?dirDB=TF_8.3 (accessed on 2 July 2021)). Based on the sequence of 5′-flanking region, different *erα* promoter deletion fragments (1057 bp, 825 bp and 555 bp) upstream of TSS were cloned. These fragments were digested with endonuclease XhoI and KPNI (New England Biolabs, Ipswich, MA, USA) and then inserted into pGL4.10 vector (Promega, Madison, WI, USA). Based on the deletion analysis, three transcription factor binding sites of *erα* promoter were mutated using Mut Express II Fast Mutagenesis Kits (Vazyme, Nanjing, China) according to the manufacturer’s protocol, and subsequently subcloned into pGL4.10 vector. In addition, the open reading frame (ORF) of *erβ2* was cloned and subcloned into pcDNA3.1. All constructs were verified by sequencing to ensure there was no mismatch. Primers used for rapid amplification, site-directed mutagenesis, and plasmid construction are listed in Table 3.

### 4.8. Dual-Luciferase Assay

Human embryonic kidney cells, (HEK293T) purchased from Shanghai Institute of Biochemistry and Cell Biology (SIBCB), were seeded on a 48-well plate and cultured in phenol red-free DMEM medium (HyClone, Logan, UT, USA) containing 5% FBS and 1% penicillin and streptomycin at 37 °C with 5% CO_2_ overnight. Cells in each well were co-transfected with 250 ng of promoter plasmid vector, 100 ng of ERβ2 expression vector, and 10 ng of pRL-TK vector (Promega). Six hours after transfection, the medium was replaced with fresh DMEM medium containing different doses of E_2_. After 24 h incubation, cells were harvested and lysed by cell lysis buffer (Vazyme). The luciferase activity was measured by the Dual-Luciferase Report Assay Kit (Promega).

### 4.9. Statistical Analysis

All statistical analyses were performed using SPSS 23.0. All data are presented as means ± SEM. Statistical differences were estimated by unpaired Student’s *t*-test or one-way ANOVA followed by Tukey’s post-hoc test. Values were considered significantly different at *p* < 0.05.

## Figures and Tables

**Figure 1 ijms-23-08632-f001:**
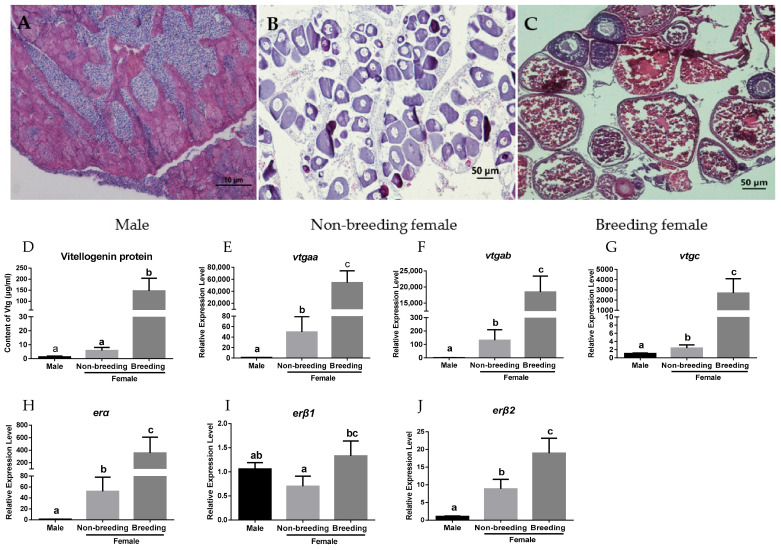
Gonad histology and expression profiles of *vtgs* and *ers* in the livers of male and female groupers: (**A**) male testis (*n* = 6) with active spermatogenesis; (**B**) non-breeding female ovaries (*n* = 6) with primary oocytes; (**C**) breeding female ovaries (*n* = 6) with vitellogenic oocytes; (**D**) VTG protein levels in the liver of males and females; (**E**–**J**) the mRNA levels of *vtgaa*, *vtgab*, *vtgc*, *erα*, *erβ1,* and *erβ2* in the liver of males and females. Data are expressed as the mean ± SEM (*n* = 6) and analyzed by one-way ANOVA followed by Tukey’s post-hoc test. Different letters above the error bars indicate statistical differences at *p* < 0.05.

**Figure 2 ijms-23-08632-f002:**
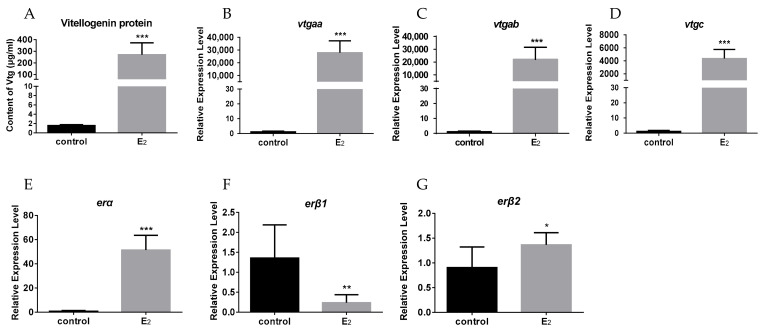
The content of VTG protein and mRNA expression of *vtgs* and *ers* in the livers of juvenile female groupers in injection group (*n* = 8) (5mg/kg E_2_) and control group (*n* = 8): (**A**) VTG protein levels in the liver; (**B**–**G**) the mRNA levels of *vtgaa*, *vtgab*, *vtgc*, *erα*, *erβ1*, and *erβ2* in the liver. Data are expressed as the mean ± SEM (*n* = 8) and analyzed by unpaired Student’s *t*-test. Asterisks (*) indicate statistical differences (* *p* < 0.05, ** *p* < 0.01, *** *p* < 0.001).

**Figure 3 ijms-23-08632-f003:**
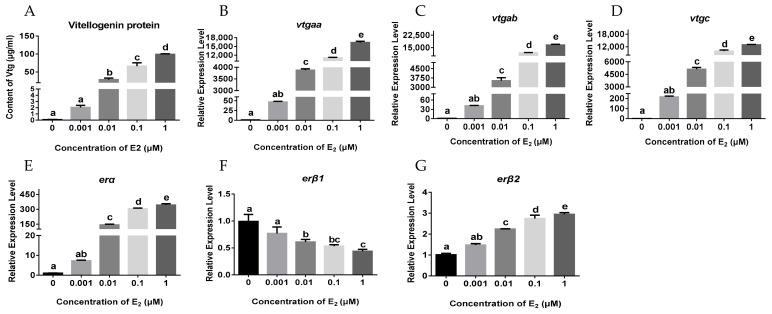
Effects of E_2_ treatment on the expression of *vtgs* and *ers* in primary hepatocytes: (**A**) VTG protein levels in the hepatocytes; (**B**–**G**) the mRNA levels of *vtgaa*, *vtgab*, *vtgc*, *erα*, *erβ1* and *erβ2* in the hepatocytes. The cells were treated with different dose of E_2_ for 24 h. Data are expressed as the mean ± SEM (*n* = 4) and analyzed by one-way ANOVA followed by Tukey’s post-hoc test. Different letters above the error bars indicate statistical differences at *p* < 0.05.

**Figure 4 ijms-23-08632-f004:**
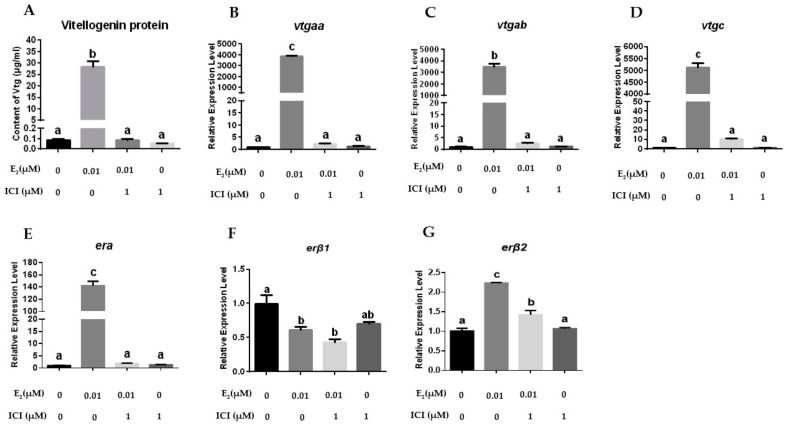
Effects of ICI182780 (1 μM) on the expression of *vtgs* and *ers* in primary hepatocytes: (**A**) VTG protein levels in the hepatocytes; (**B**–**G**) the mRNA levels of *vtgaa*, *vtgab*, *vtgc*, *erα*, *erβ1* and *erβ2* in the hepatocytes. Data are expressed as the mean ± SEM (*n* = 4) and analyzed by one-way ANOVA followed by Tukey’s post-hoc test. Different letters above the error bars indicate statistical differences at *p* < 0.05.

**Figure 5 ijms-23-08632-f005:**
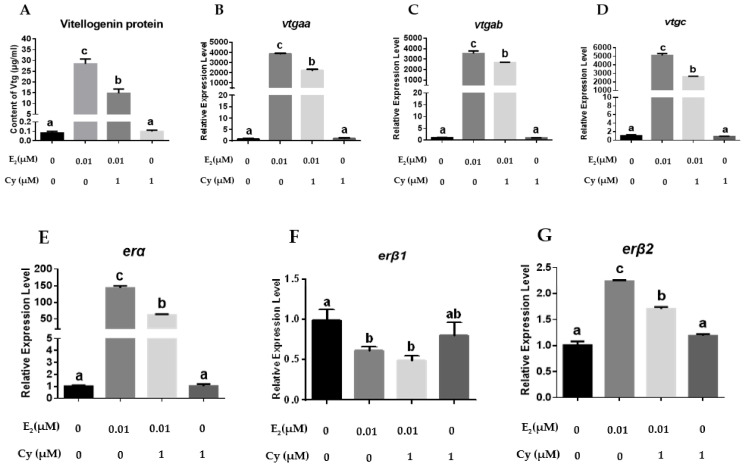
Effects of Cyclofenil (1 μM) on the expression of *vtgs* and *ers* in primary hepatocytes: (**A**) VTG protein levels in the hepatocytes; (**B**–**G**) the mRNA levels of *vtgaa*, *vtgab*, *vtgc*, *erα*, *erβ1* and *erβ2* in the hepatocytes. Data are expressed as the mean ± SEM (*n* = 4) and analyzed by one-way ANOVA followed by Tukey’s post-hoc test. Different letters above the error bars indicate statistical differences at *p* < 0.05.

**Figure 6 ijms-23-08632-f006:**
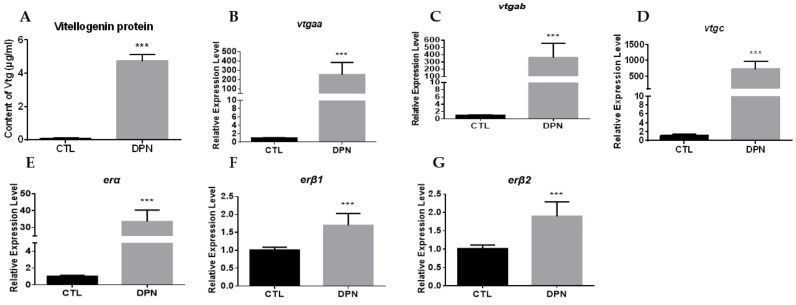
Effects of DPN (1 μM) on the expression of *vtgs* and *ers* in primary hepatocytes: (**A**) VTG protein levels in the hepatocytes; (**B**–**G**) the mRNA levels of *vtgaa*, *vtgab*, *vtgc*, *erα*, *erβ1* and *erβ2* in the hepatocytes. Data are expressed as the mean ± SEM (*n* = 4) and analyzed by unpaired Student’s *t*-test. Asterisks (*) indicate statistical differences (*** *p* < 0.001).

**Figure 7 ijms-23-08632-f007:**
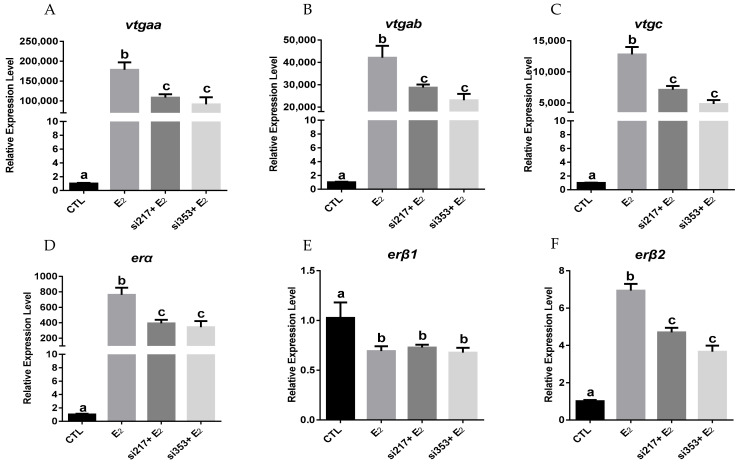
Effects of siRNA knockdown on the expression of *vtgs* and *ers* in primary hepatocytes: (**A**–**F**) the mRNA levels of *vtgaa*, *vtgab*, *vtgc*, *erα*, *erβ1* and *erβ2* in the hepatocytes. The cells were transfected with si217 or si353 (5 μg /mL), and then treated with 0.01 μM E_2_ for 24 h. Data are expressed as the mean ± SEM (*n* = 4) and analyzed by one-way ANOVA followed by Tukey’s post-hoc test. Different letters above the error bars indicate statistical differences at *p* < 0.05.

**Figure 8 ijms-23-08632-f008:**
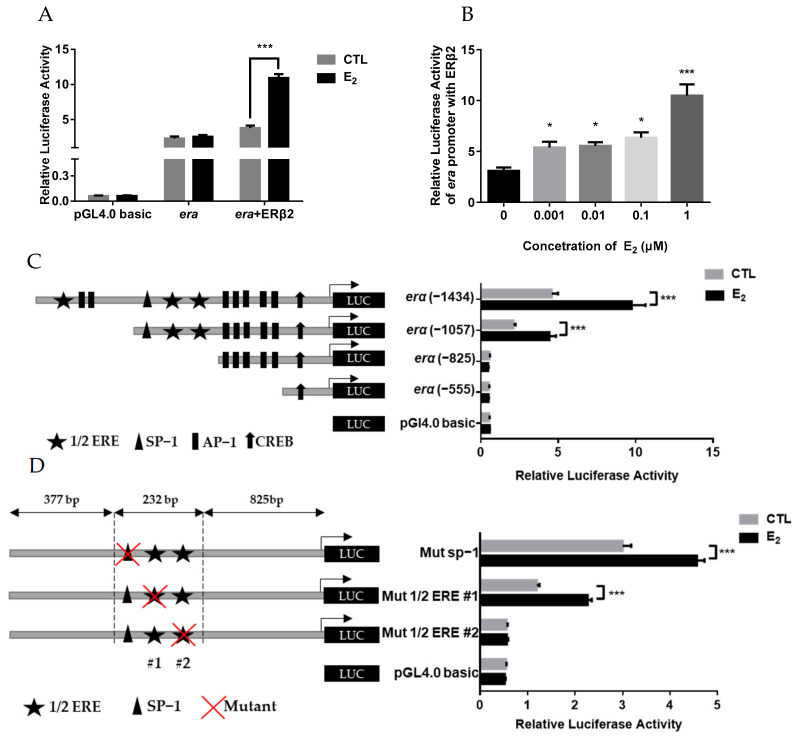
The luciferase activities of *erα* promoter in HEK293T cells. Cells in each well were co-transfected with 250 ng of promoter plasmid vector, 100 ng of ERβ2 expression vector, and 10 ng of pRL-TK vector (Promega). (**A**) Basal activities of *erα* promoter in transfected HEK293T cells. Transfected cells were treated with ethanol (CTL) or 1 μM E_2_ for 24 h. (**B**) The luciferase activities of *erα* promoter after different dose of E_2_ treatment in the presence of ERβ2. (**C**) Deletion analysis of *erα* promoter. Left panel: schematic representation of *erα* promoter deletion constructs and the positions of the putative motifs. Right panel: the luciferase activities of corresponding *erα* promoter deletion constructs. 293T cells. Transfected cells were treated with ethanol (CTL) or 1 μM E_2_ for 24 h. (**D**) Mutation analysis of *erα* promoter. Left panel: schematic representation of *erα* promoter mutant constructs. Right panel: the luciferase activities of corresponding *erα* promoter mutant constructs. Transfected cells were treated with ethanol (CTL) or 1 μM of E_2_ for 24 h. Data were expressed as the mean ± SEM (*n* = 4) of the relative luminescence activity from each promoter construct and analyzed by unpaired Student’s *t*-test (**A**,**C**,**D**) or one-way ANOVA followed by Tukey’s post-hoc test (**B**). * *p* < 0.05; *** *p* < 0.001 compared to the corresponding control.

**Table 1 ijms-23-08632-t001:** siRNA used in knockdown experiment.

Gene	Gene Accession	siRNA (Sense Strand)
*erβ2*	GU721078.1	GCCCAUCUGUGCUGAGCUATT
*erβ2*	GU721078.1	GCCUCUCGUCUACAAUGAATT
NC		UUGUCCGAACGUGUCACGUTT

**Table 2 ijms-23-08632-t002:** Primers used for Real-Time PCR.

Gene	Primer Name	5′ to 3′ Sequence
*vtgaa*	*vtgaa*-F	GTATCCCAACAAGTTCCAGAGG
	*vtgaa*-R	GGACGATGATGGCAAAGGTAG
*vtgab*	*vtgab*-F	GCTGCCCGCCTGAAGATTAC
	*vtgab*-R	CCTTTGCCAGGTTTATTTCG
*vtgc*	*vtgc*-F	CTGCGAGCAATGCCTTAT
	*vtgc*-R	GGAATGGCCTTGAGATGG
*erα*	*erα-*F	GGACACCATCACAGATGCTCTC
	*erα-*R	CTCTGTTTGGGCTCTGGTGGCTG
*erβ1*	*erβ1-*F	GACAAGAACCGCCGTAAGAGC
	*erβ1-*R	GAGAAGATAAGTTTCCCTGGATG
*erβ2*	*erβ2-*F	TCACCAACCTGGCAGACAAGGAG
	*erβ2-*R	GTACACAGATTGTAGTTAAGGAG
*beta-actin*	*actin-*F	ACCATCGGCAATGAGAGGTT
	*actin-*R	ACATCTGCTGGAAGGTGGAC

**Table 3 ijms-23-08632-t003:** Primers used for rapid amplification, site-directed mutagenesis, and plasmid construction.

Gene	Primer Name	5′ to 3′ Sequence
Primers used for rapid amplification of cDNA ends		
*erα*	*erα*-race-R1	AGTCATTGTGACCCTGAATGCTCC
	*erα*-race-R2	CATACTGTATGCCTCGTCACTG
Primers used for Site-directed mutagenesis		
*erα*	Mut-sp1-F	CCACCtcgataaagCATTTGGGATTGATTGTGTAATTATTG
	Mut-sp1-R	ATGctttatcgaGGTGGCTATTTTAATCAGATGCTG
*erα*	Mut-ERE1-F	GAGATTgacttcCCTGCTTTGTTTGCTGTGTTATG
	Mut-ERE1-R	AGCAGGgaagtcAATCTCATTCCAACAGCCAATAATT
*erα*	Mut-ERE2-F	TTATGGgacttcGGCAGTAAAACACACTGCTGTTTG
	Mut-ERE2-R	ACTGCCgaagtcCCATAACACAGCAAACAAAGCAG
Primers used for plasmid construction		
*erα*	*erα*-1434-F	TGGCCTAACTGGCCGGTACCGAGCTCCTGTTGTAACTGGT
	*erα*-1434-R	TCTTGATATCCTCGAGGTCACTGAAGGGGGCACGA
	*erα*-1057-F	TGGCCTAACTGGCCGGTACCCAATGAAATGTCATGAGG
	*erα*-825-F	TGGCCTAACTGGCCGGTACCGCATCTTTAATTTGTTTATC
	*erα*-555-F	TGGCCTAACTGGCCGGTACCTTACTGCAGAGTCTCAGG
*erβ2*	*erβ2*-ORF-F	CTAGCTAGCATGGCCTCGTCCCCTGAGCT
	*erβ2*-ORF-R	CCGGAATTCCTACTGGTTCCACTGATGGA

## Data Availability

The data underlying this article will be shared on reasonable request to the corresponding author.

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
