# Peer review of "Distinct Roles of Estrogen Receptors in the Regulation of Vitellogenin Expression in Orange-Spotted Grouper (Epinephelus coioides)"

_ijms, 2022, doi:10.3390/ijms23158632_

Round 1

Reviewer 1 Report

In this paper the authors have taken a close look at the regulation of vitellogenin production in the liver of orange-spotted grouper, a commercially important teleost fish. They are interested specifically in regulation by estradiol and finding out which estrogen receptor subtypes are the most important. They have conducted a series of elegant studies. First, they looked at how vitellogenin production and mRNA expression of three different subtypes of vitellogenin as well as mRNA expression of three types of ER differ seasonally (breeding vs. non-breeding). This study showed that all of these dependent variables increase significantly, in some cases dramatically, between non-breeding and breeding females.

Second, they showed that treatment with estradiol increased all of the same variables except ERB1 mRNA, which dramatically decreased – a paradoxical finding when compared with the results of Study #1. Only one dose of E2 was used, but the authors followed up by looking at the dose-dependent effects of E2 in cultured hepatocytes. All of the findings were replicated, including the inhibitory effect of E2 on ERB1 mRNA expression.

Next, the authors tested the effects of several ER agonists and antagonists in vitro. The first antagonist blocked all of the previously shown effects except the inhibitory effect on ERB1. A specific ERb antagonist attenuated but did not completely block the effects of E2 on all variables except, again, ERB1mRNA. An ERb-specific agonist increased all of the variables, even expression of ERB1. Finally, knockdown of ERB2 mRNA attenuated but did not block responses to E2 except, again, the inhibitory effect on ERB1.

The authors then conducted reporter assays in HEK cells to test the ability of E2 to stimulate transcription of ERa in the presence and absence of ERB2 (presumably in the absence of ERa and ERB1). Deletion analyses identified a particular region of the era promoter that responds to E2.

Overall the studies are well-designed and easy to understand. Although the manuscript needs editing for English usage, I had no trouble following the writing. I had a number of concerns mostly related to the omission of important information like sample sizes and other details of methods, as outlined below.

General comments:

As the authors quantified the expression of a number of different genes in the same samples (e.g. Fig. 1, Fig. 2) please make sure to correct for multiple comparisons by adjusting p values or alpha values in the one-way ANOVAs.

The results need more exposition/explanation as you proceed from step to step. Please add more rationale for each step at the beginning of each section. For example, what was the rationale for testing these particular agonists and antagonists? What was the rationale for testing only the era promoter, and only ERB2 protein, in the reporter assays? Why not, for example, ERB1 promoter, since this gene is perhaps one of the most interesting from the previous studies? It will help the reader a lot to see your logic. Some of the information from Lines 312-339 could be pulled into the Results to help the reader understand the logical progression from study to study.

Although I enjoyed reading this manuscript as a basic scientist, the significance of the findings are placed in the context of fisheries, making me wonder whether this is the right journal and whether the manuscript is more suited to a more applied journal.

Although the figures are clear and elegant, the font size is too small for me.

More specific comments:

Introduction

Lines 44-68 (also lines 310-311 in the Discussion): given the convincing evidence from other species that vitellogenin production is regulated by ER, and even by particular ERs, I think the authors could make a stronger case here for why their current study is novel and necessary. What in particular is the precise gap that the study will fill, other than being in a different species?

Results

L90, it’s a little unclear how males serve as a “control”. What is being controlled? Is it that they are not engaged in egg production? It looks like they were collected at the same time as the breeding females, with the non-breeding females collected at a different time, so the design is slightly confounded. It is interesting that there is no sex difference for ERB1 but the significance of that finding is not quite clear given that the use of males as a control is not well-explained.

Relatedly, on Line 117 the “control group” changes to something else but it’s not specified what it is. The sex of the animals in Fig. 2 is not given. Please indicate sex in the captions of all figures and specify at the beginning of section 2.2 (line 116) that this study was conducted in females only, if that is the case.

Figures 1 and 2: in the caption or in the figures please give sample sizes for each group.

Line 136 please specify here the sex and breeding condition of the animals from which the hepatocytes were dissected.

Line 139-140, the text is backwards when referring to Figures 3F and 3G.

Line 159, please indicate which ER this antagonist is thought to block. If it is nonspecific please indicate such.

Section 2.5, e.g. Line 255, please comment on the availability of E2 in the media as well as ERa, ERB1 and ERB2 in these cells.

Line 290, please explain the ERa + ERB2 condition in the caption.

Lines 312-322, since this finding is among the most interesting in the study it would be nice to see a little more consideration here about the mechanisms potentially underlying this inhibitory effect and more information about what is known about regulation of ERB1.

Lines 382 and 402, why is there such a large age difference between the first and second study? Was the body weight in the IP injection study comparable to the others? What was the source of the animals in the IP injection study? L404, what is meant by ‘domesticated’? were they collected from the wild?

Line 382 note spelling of “weight”

Line 382 please give the sample sizes.

Line 404, please provide a rationale for the dosage. Why was this particular dosage chosen?

Section 4.4, we need more details about the cultures and this analysis. How many females were used? Did each culture contain cells from only one female or where they mixed individuals? How many replicates (wells/plates) were used per drug, were the data normalized to a particular replicate, etc.? And were all of the genes and vitellogenin measured in the same samples, e.g. in Figs. 3-6 or were separate cultures done for each gene? If the former, please see my concern above about correcting statistically for multiple comparisons.

L 420, please specify whether this medium contains steroid hormones and if so which ones and the concentrations.

L460, please provide a reference showing validation of Beta-actin as a reference gene in this species and justify the choice of using just one reference gene instead of a combination.

Reviewer 2 Report

This study has revealed the distinct roles of estrogen receptors in the regulation of vitellogenin expression in orange-spotted grouper. The evidence is solid and can well support the conclusion. Also this manuscript was well written. I think this manuscript can be accepted for publication in International Journal of Molecular Sciences with the minor modifications listed below.

1. line 21: Add “ER antagonist” before “ICI182780”.

2. line 27: Change “half-ERE” into “half-estrogen response elements (ERE)”.

3. line 85: Change “era” into “erα”.

4. line 352: Delete the word “directly”.

5. line 369: Change “knockout down” into “knockdown”.

6. line 378: The words “erα” and “vtg” should be in italic.
